# Structure and antigenicity of the divergent human astrovirus VA1 capsid spike

**Anisa Ghosh**[1], **Kevin Delgado-Cunningham**[1], **Tomás López**[2], **Kassidy Green**[1], **Carlos F. Arias**[2], **Rebecca M. DuBois**[1] *

**1** Department of Biomolecular Engineering, University of California Santa Cruz, Santa Cruz, California, United States of America, **2** Departamento de Genética del Desarrollo y Fisiología Molecular, Instituto de Biotecnología, Universidad Nacional Autónoma de México, Cuernavaca, Morelos, Mexico

* rmdubois@ucsc.edu

**Data Availability Statement:** The coordinates and structure factors for the gastrointestinal and neuronal HAstV-VA1 capsid spike X-ray crystal structures have been deposited with the Protein Data Bank (PDB; https://www.rcsb.org) as entry 8UFO and 8UFN.

## Abstract

Human astrovirus (HAstV) is a known cause of viral gastroenteritis in children worldwide, but HAstV can cause also severe and systemic infections in immunocompromised patients. There are three clades of HAstV: classical, MLB, and VA/HMO. While all three clades are found in gastrointestinal samples, HAstV-VA/HMO is the main clade associated with meningitis and encephalitis in immunocompromised patients. To understand how the HAstV-VA/HMO can infect the central nervous system, we investigated its sequence-divergent capsid spike, which functions in cell attachment and may influence viral tropism. Here we report the high-resolution crystal structures of the HAstV-VA1 capsid spike from strains isolated from patients with gastrointestinal and neuronal disease. The HAstV-VA1 spike forms a dimer and shares a core beta-barrel structure with other astrovirus capsid spikes but is otherwise strikingly different, suggesting that HAstV-VA1 may utilize a different cell receptor, and an infection competition assay supports this hypothesis. Furthermore, by mapping the capsid protease cleavage site onto the structure, the maturation and assembly of the HAstV-VA1 capsid is revealed. Finally, comparison of gastrointestinal and neuronal HAstV-VA1 sequences, structures, and antigenicity suggests that neuronal HAstV-VA1 strains may have acquired immune escape mutations. Overall, our studies on the HAstV-VA1 capsid spike lay a foundation to further investigate the biology of HAstV-VA/HMO and to develop vaccines and therapeutics targeting it.

## Author summary

Human astroviruses, well-established as a leading cause of viral diarrhea, are increasingly associated with viral encephalitis in immunocompromised patients, with the sequence-divergent VA1 strains being the most common in these cases. Our study presents the high-resolution capsid spike structures from VA1 strains isolated from patients with both gastrointestinal and neuronal manifestations. Compared to the classical human astrovirus capsid spike structure, the VA1 capsid spike structure has striking differences in size, shape, and surface features, suggesting that the divergent VA1 strains may have evolved a different mechanism to attach to host cells, which in turn may influence their ability to

**Funding:** This research was supported by NIH/NIAID grant R01AI144090 to R.M.D. and C.F.A., and grant CONACyT M0037-Fordecyt 302965 to C. F.A. The funders had no role in the study design, data collection and analysis, decision to publish, or the preparation of the manuscript.

**Competing interests:** The authors have declared that no competing interests exist.

infect the central nervous system. On the other hand, comparison of the gastrointestinal and neuronal VA1 spike structures reveals few differences. Instead, sequence and antigenic studies suggest that neuronal VA1 strains isolated from immunocompromised patients may have acquired mutations to escape immunoglobulin therapy. Altogether, this work provides a structural basis to investigate the mechanism of infection by the divergent human astrovirus VA1 strains and to develop tailored diagnostics, vaccines, and therapeutics against them.

## Introduction

Astroviruses, first described in 1975 and belonging to the *Astroviridae* family, are non-enveloped, positive-sense single-stranded RNA viruses known to infect a wide range of avian and mammalian species [1]. Classical human astroviruses (HAstVs) include serotypes 1–8, and are associated with acute or severe gastroenteritis mainly in children; almost 90% of children possess detectable antibodies to one of the classical HAstV serotypes [2–5]. During the last decade, two clades of novel and highly divergent human astroviruses, HAstV-MLB and HAstV-VA/HMO, were discovered in patients suffering from gastroenteritis [6–10]. However, a definitive association between these novel clades and gastroenteritis has not yet been established [11,12]. Phylogenetic studies of ORF2-encoded capsid protein sequences showed that these divergent MLB and VA/HMO clades are more closely related to animal astroviruses than to classical HAstVs. For example, HAstV-VAs are most closely related to ovine and mink astroviruses, resulting in the naming of the Human Mink Ovine (HMO) clade [13–20]. Notably, many of these related animal astroviruses are found to be associated with neurological disease in ovine, minks, bovine, porcine, alpacas, and muskox [13,20,21], and HAstV-VAs are increasingly being associated with neurological disease. In 2010, HAstV-VA1 was first identified by metagenomic sequencing as the causative agent of encephalitis in an immunocompromised patient with X-linked agammaglobulinemia who ultimately died after 71 days of hospitalization [6]. To date, 15 cases of neuronal HAstV infections have been reported in immunocompromised individuals, causing encephalitis or meningitis and resulting in high mortality rates [22–34]. Of these cases, a majority (9/15) were caused by HAstV-VA1. Recently, a study of adult and pediatric serum samples showed a high seroprevalence of neutralizing antibodies to HAstV-VA1, with a seropositivity rate of 77% in adults [35]. Altogether, these studies demonstrate that humans are commonly exposed to HAstV-VA1, and rare but often fatal cases of HAstV infections in the central nervous system (CNS) occur in immunocompromised individuals.

HAstVs have an icosahedral capsid that encapsulates the ~6.8 kb positive-sense ssRNA genome. The genome of astrovirus is comprised of 5′ and 3′ untranslated regions and four open reading frames (ORFs) (ORF1a, ORF1b, ORFX, and ORF2). The non-structural polyproteins nsp1a and nsp1ab, encoded by ORF1a and ORF1b, are translated from the genomic RNA; these proteins are proteolytically processed into smaller proteins to yield the RNA-dependent RNA polymerase, a serine protease, a viral genome-linked protein (VPg), and several other proteins with unknown functions [36–39]. The other two ORFs, ORFX and ORF2, are overlapping and translated from the subgenomic RNA. ORFX, which is found in only genogroup I Astroviruses, encodes a viroporin protein [40,41]. ORF2 encodes the capsid precursor protein VP90 [37–39].

The HAstV capsid precursor protein comprises several domains, including a highly basic amino terminus that binds to the RNA genome, a core domain that forms the icosahedral

shell, a spike domain that forms dimeric protrusions, and an acidic domain with unknown function at the carboxy terminus (Fig 1B). In classical HAstVs, the capsid precursor protein VP90 self-assembles and is cleaved by intracellular host caspases to remove the acidic domain, resulting in VP70 immature HAstV particles [42–44]. After release from the infected cells, further proteolytic processing by host extracellular proteases yields mature HAstVs. *In vitro*, trypsin is required to generate the mature, infectious virus for the classical HAstVs, resulting in two predominant capsid proteins: VP34 and VP27 [42,43,45,46]. In contrast, the HAstV-VA1 capsid precursor protein VP86 is cleaved intracellularly in a caspase-independent manner by one or more unknown proteases [47].

Structural and mechanistic studies have shed light on the HAstV capsid roles in virus entry. First, cryoelectron microscopy was used to elucidate the structure of mature classical HAstV, revealing a ~35nm diameter T = 3 icosahedral capsid studded with dimeric spike protrusions

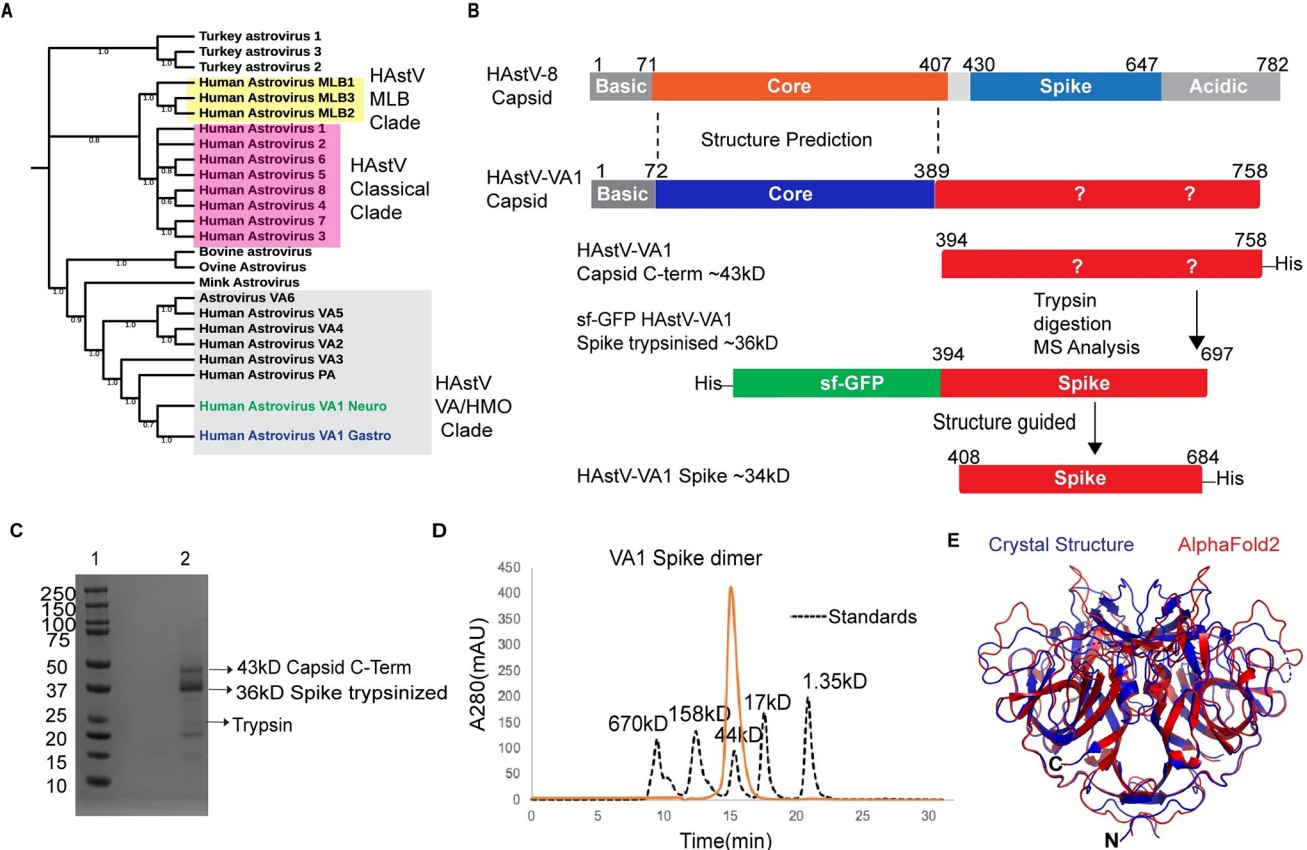

**Fig 1. Phylogenetic analysis of HAstVs and delineation of the HAstV-VA1 capsid spike domain.** (A) Phylogenetic analysis of human and animal astroviruses. Complete capsid sequences were aligned using the MUSCLE Algorithm, and evolutionary analysis was done using MEGA 11 maximum Likelihood and JTT matrix-based model to yield the cladogram shown. Bootstrap values are shown next to the branches and were computed using 1,000 replicates. Turkey astrovirus 1–3 capsid sequences were used as an outgroup. (B) Schematic of HAstV-8 and HAstV-VA1 capsid structural domains and recombinant HAstV-VA1 capsid protein constructs. Question marks indicate the unknown termini of the spike domain and acidic region. (C) Coomassie-stained SDS-PAGE of limited proteolysis of HAstV-VA1 Capsid C-term with trypsin. Lane1: molecular weight marker, in kD (Biorad Precision Plus Protein Dual Color Standards). Lane 2: trypsin digestion products showing bands for the 43 kD capsid C-term, the 36 kD Spike trypsinized, and trypsin. (D) Superdex 200 size-exclusion chromatography trace of recombinant HAstV-VA1 spike dimer (orange) overlaid with the trace of gel filtration standards (black dotted line). (E) Structural alignment between the 1.46 Å-resolution crystal structure of HAstV-VA1 spike and the AlphaFold2-predicted HAstV-VA1 spike, which was predicted for the 15th Community Wide Experiment on the Critical Assessment of Techniques for Protein Structure Prediction (CASP15). A structural alignment was performed using TM ALIGN software with a TM-Score of 0.85 and an RMSD of 2.90 Å across 264 residue pairs.

at the icosahedral 2-fold axes [48]. Subsequent X-ray crystallographic studies elucidated the high-resolution structures of the HAstV-1 and -8 capsid core domains [49,50], as well as the capsid spike domains from HAstV-1, -2, and 8, turkey astrovirus 2, and the novel divergent HAstV-MLB1 [51–55]. The spike domain is utilized by classical HAstVs for attachment to host cells, and neutralizing antibodies that block classical HAstV attachment to human Caco-2 cells bind to several distinct epitopes on the spike domain [54,56,57]. Previous studies mapped putative receptor-binding sites onto the classical HAstV spike structures [52]. Recently, the neonatal Fc receptor (FcRn) was identified as a candidate receptor for classical HAstVs [58]. Notably, the classical HAstV-1, -2, and -8 spike structures are structurally similar (RMSD ~1.2 Å), and while the HAstV-MLB1 spike structure retains a similar overall fold, it is more structurally divergent (RMSD ~3.7 Å) and does not have any of the putative receptor-binding sites found on classical HAstV spikes [51,52]. The structure of the divergent HAstV-VA spike has remained elusive.

Here we report the high-resolution crystal structures of the capsid spike of two HAstV-VA1 strains: one strain isolated from a gastrointestinal infection, and a second obtained from a brain biopsy of a patient with neurological disease. These structures provide insights into HAstV-VA entry, evolution, maturation, and antigenicity. Specifically, comparison of the HAstV-VA1 spike structure with classical HAstV and HAstV-MLB spike structures reveals a related dimer formation and core beta-barrel structure. However, the HAstV-VA1 spike is otherwise larger and strikingly different, suggesting that it may utilize a different mechanism to attach to host cells. N-terminal sequencing and mass spectrometry analyses of mature HAstV-VA1 revealed the cleavage site of the capsid precursor VP86 protein, illuminating the maturation and assembly of the HAstV-VA1 capsid VP33 and VP38 proteins in the context of the capsid structural domains. Finally, antigenic studies support that neuronal HAstV-VA1 strains have acquired immune escape mutations. Overall, these studies provide a structural basis to understand HAstV-VA/HMO viruses and support the development of vaccines and therapeutics against these divergent HAstVs.

## Results

### Delineation of the HAstV-VA1 capsid spike domain

To delineate the HAstV-VA capsid spike structural domain, we first evaluated the evolutionary relationship of complete capsid protein sequences from classical, MLB, and VA/HMO clades, as well as related animal astroviruses (Fig 1A). We began by studying the capsid sequence from the first neuronal HAstV-VA1 (HAstV-VA1[neuro]) found to be associated with encephalitis in an immunocompromised patient who ultimately died after 71 days of hospitalization [28]. Sequence alignment of this HAstV-VA1[neuro] capsid protein sequence with that of classical HAstV-8 identified a region in the first ~400 amino acids of each protein with ~40% sequence identity, and HHPred structural analysis predicted structural homology to the capsid core structural domain in HAstV-1 and -8 [49] (Fig 1B). In contrast, there was no sequence identity in the C-terminal regions of the capsid proteins that form the spike structural domain and the acidic region in HAstV-8 (Fig 1B). We first generated over 15 constructs with varying N- and C-termini that we predicted would form the HAstV-VA1 capsid spike structural domain, however all attempts to express these proteins recombinantly in *E. coli* resulted in insoluble protein. In a different approach, we generated a larger construct encoding the C-terminal residues 394–758 of the HAstV-VA1[neuro] capsid protein (~43 kD) (Fig 1B), and expressed this protein in Sf9 insect cells. Despite low yields (micrograms per Liter), this recombinant protein was soluble, and limited proteolysis with trypsin resulted in a trypsin-resistant fragment of ~36 kD (Fig 1C). Mass spectrometry analysis revealed a mass of 36,020 Daltons,

**Table 1. Crystallography data collection and refinement statistics.**

|  | VA1 Spike$^{Gastro}$ | VA1 Spike$^{Neuro}$ |
|---|---|---|
| **Data Collection**[a] |  |  |
| PDB Code | 8UFO | 8UFN |
| Space group | C 1 2 1 | P $2_1$ $2_1$ $2_1$ |
| a, b, c (Å) | 109.62, 82.56, 61.12 | 61.203, 86.32, 108.85 |
| α, β, γ (°) | 90, 102.71, 90 | 90, 90, 90 |
| Resolution (Å) | 65.35–1.46 (1.49–1.46) | 46.04–2.73 (2.86–2.73) |
| $R_{merge}$ | 0.051 (0.150) | 0.354 (2.102) |
| $R_{pim}$ | 0.019 (0.065) | 0.144 (0.876) |
| I/σI | 18.7 (7.2) | 7.6 (1.5) |
| Completeness (%) | 100.0 (99.9) | 100.0 (100.0) |
| Multiplicity | 6.4 (5.4) | 13.0 (12.6) |
| $CC_{1/2}$ | 0.999 (0.981) | 0.989 (0.616) |
| **Refinement** |  |  |
| No. of reflections | 91236 (9043) | 15876 (1541) |
| Resolution (Å) | 41.28–1.46 | 46.04–2.73 |
| $R_{work}$/$R_{free}$[b] | 0.152 / 0.176 | 0.251 / 0.310 |
| Atoms | 4880 | 4279 |
| Protein | 4322 | 4279 |
| Water | 558 | 0 |
| Mean B factor (Å$^2$) | 14.82 | 46.44 |
| Protein (Å$^2$) | 13.44 | 46.44 |
| Water (Å$^2$) | 25.55 | N/A |
| RMSD |  |  |
| Bond lengths (Å) | 0.007 | 0.003 |
| Bond angles (°) | 1.01 | 0.59 |
| Ramachandran statistics |  |  |
| Favored (%) | 99.06 | 91.03 |
| Allowed (%) | 0.94 | 8.9 |
| Outliers (%) | 0.00 | 0.00 |

[a] The values in parentheses are for the outermost shell.

[b] $R_{free}$ is the $R_{work}$ based on 10% of the data excluded from the refinement.

similar to a predicted tryptic fragment of 35,970 Daltons that would be generated by cleavage after arginine 697. Thus, a new construct encoding the HAstV-VA1$^{neuro}$ capsid residues 394–697 (~36 kD) was generated as a fusion protein with superfolder GFP (sf-GFP) (Fig 1B) and expressed in insect cells with improved yields (milligrams per Liter). Following removal of the sf-GFP, the HAstV-VA1$^{neuro}$ capsid spike protein was crystallized and diffraction data to a resolution of 2.73 Å was collected (Table 1).

## Crystal structure determination of the HAstV-VA1 capsid spike

At the time, no predicted structural model allowed us to successfully obtain a solution by molecular replacement. A predicted structural model generated by AlphaFold2 for the 2022 CASP15 competition also did not work. However, we gradually deleted amino acids with the lowest pLDDT confidence scores (ultimately removing ~20% of the model's amino acids with pLDDT scores <55) and finally obtained a convincing molecular replacement solution.

The structure of the HAstV-VA1[neuro] capsid spike was solved to 2.73 Å resolution (Table 1 and S1 Fig). Structural alignment of the experimentally-determined HAstV-VA1 spike crystal structure with the Alphafold2 predicted model using TM-align revealed an RMSD of 2.90 Å across 264 residue pairs and a TM-score of 0.85, revealing a relatively accurate match in the beta-strand and alpha-helical regions and significant differences in the loop regions (Fig 1E).

The HAstV-VA1[neuro] capsid spike structure provided a guide to generate a new construct encoding only the amino acids that form the HAstV-VA1 spike structural domain (residues 408–684) (Fig 1B), and this construct yielded a soluble protein in *E. coli*. We then used this strategy to generate a construct encoding the capsid spike structural domain from a gastrointestinal HAstV-VA1 (HAstV-VA1[gastro]) identified in 2008 in an individual with acute gastroenteritis [6]. This recombinant HAstV-VA1[gastro] capsid spike protein forms a dimer in solution, consistent with other HAstV capsid spikes (Fig 1D). The HAstV-VA1[gastro] capsid spike was then crystallized and its structure was solved to 1.46 Å resolution (Table 1).

## Structural comparisons of the HAstV-VA1 spike to classical and MLB HAstV spikes

The HAstV-VA1 spike structural domain forms a homodimer, with each protomer encompassing residues 408–684 (Fig 2A). We note here that our use of the terminology "HAstV-VA1 spike domain" refers to the structural domain of a single protomer, whereas "HAstV-VA1 spike" refers to the globular dimer formed by two spike domains. The HAstV-VA1 spike has ~3648 Å$^2$ buried at the dimer interface mediated by 93 interface residues, which is similar to the classical HAstV-1, -2, and -8 spikes, which have dimer interfaces of 3500–3800 Å$^2$, and also to the divergent HAstV-MLB1 spike, which has ~3100 Å$^2$ dimer interface [49,51,52,54,55] (Fig 2). Also similar to other HAstV spikes, the structural domain of each protomer of the HAstV-VA1 spike comprises a core antiparallel beta-barrel, formed by beta-strands 1, 9, 10, 16, and 18. Despite these general similarities, the overall shape, size, and surface of the HAstV-VA1 spike is strikingly different (Fig 2), with the HAstV-VA1 spike dimer being ~64kD compared to the ~50kD canonical and MLB1 spike dimers. Structural alignment of the HAstV-VA1 spike domain with HAstV-2 and HAstV-MLB spike domains using TM-align revealed RMSDs of 3.88 Å and 3.67 Å and TM-scores of 0.74 and 0.73, respectively, revealing the divergence of the structures. One notable structural difference of the HAstV-VA1 spike is the orientation of long opposing loops that extend over the top of the spike (the region between strands β12 and β14 in the HAstV-VA1 spike). In classical HAstV and HAstV-MLB spikes, the loops lie side-by-side across the top of the spike, whereas in HAstV-VA1 spike the loops wrap around each other intimately in a "yin and yang" fashion (Fig 2). Interestingly, these loops at the top of the classical HAstV spike are involved in dimerization and reactivity with neutralizing antibodies [52,57]. Altogether, these size and structural differences result in a completely different biochemical surface on the HAstV-VA1 spike compared to the classical and MLB spikes, with no obvious three-dimensional patches of sequence similarity that might indicate a conserved receptor-binding site. In other words, our structural observations support the hypothesis that classical and HAstV-VA1 may utilize a different host cell receptor.

To test this hypothesis, an infection competition assay was conducted to assess the ability of recombinant HAstV-1 or HAstV-VA1 capsid spike to impede infection by the homologous or heterologous HAstV [51]. The rationale underlying this investigation is based on the assumption that if all HAstVs share the same receptor, either spike will compete for the interaction with a common receptor, hindering infection of both the homologous and the heterologous

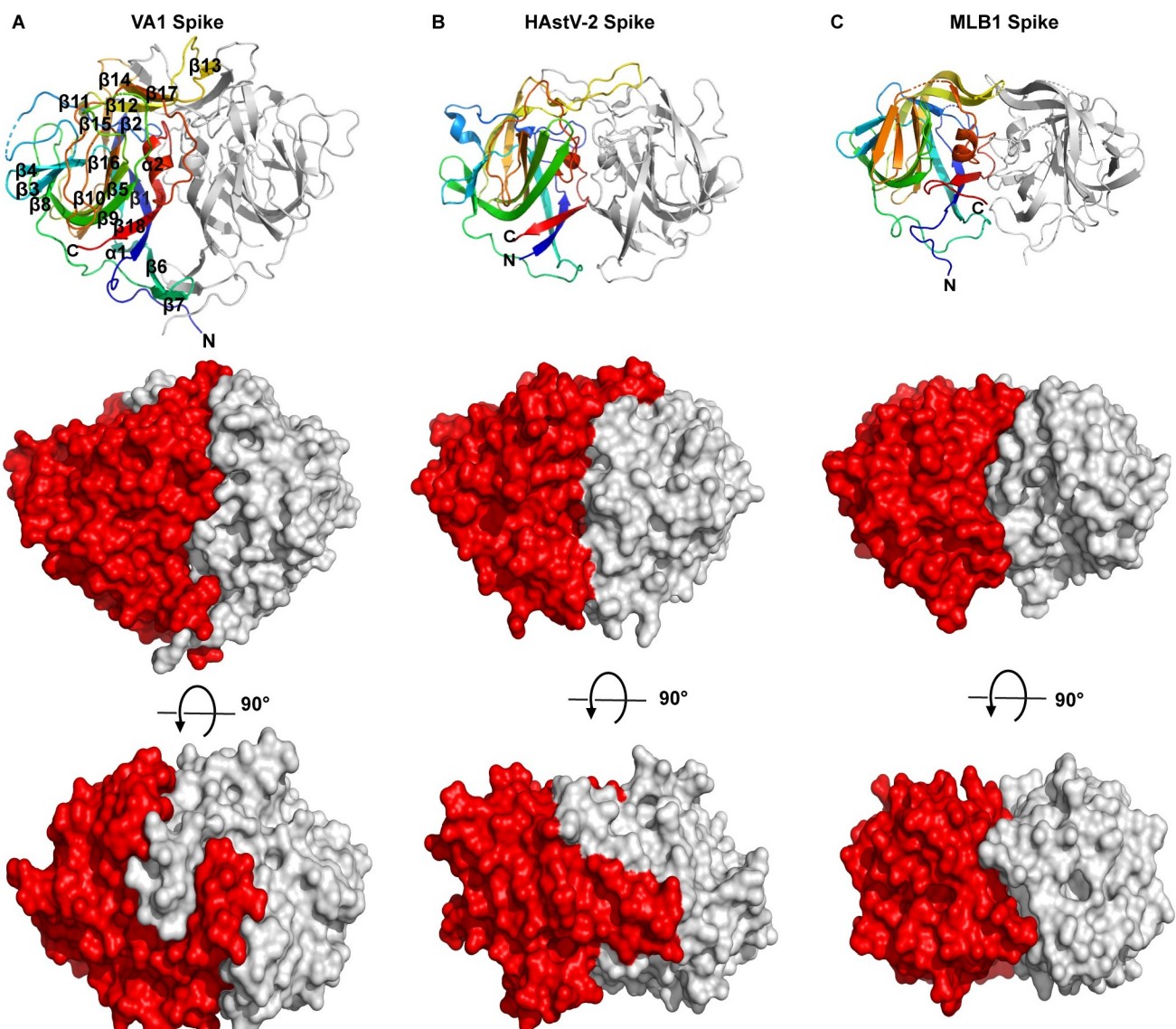

**Fig 2. Comparison of the HAstV-VA1 spike structure to the classical and MLB HAstV spike structures.** (A) Structure of HAstV-VA1 spike dimer presented as a cartoon model with labeled features on one protomer colored rainbow from the N-terminus (blue) to the C-terminus (red) (top panel). Below, the dimer is presented as a surface model (red and grey) as the side view (middle panel) and top view (bottom panel). (B) Structure of classical HAstV-2 spike dimer (PDB: 5W1N) presented as a cartoon model and colored rainbow (top panel) and presented as a surface model (middle and bottom panels). (C) Structure of HAstV-MLB spike dimer (PDB: 7UZT) presented as a cartoon model and colored rainbow (top panel) and presented as a surface model (middle and bottom panels). Flexible residues that were not visible in each structure are drawn as dashed lines.

HAstV. First, we found that recombinant HAstV-1 spike inhibited HAstV-1 infection in a dose-dependent manner, consistent with previous observations (Fig 3) [51]. Likewise, the recombinant HAstV-VA1 spike inhibited HAstV-VA1 infection in a dose-dependent manner (Fig 3). In contrast, recombinant HAstV-1 spike did not inhibit HAstV-VA1 infection, and recombinant HAstV-VA1 spike did not inhibit HAstV-1 infection (Fig 3). Thus, these data support the hypothesis that HAstV-VA1 utilizes a distinct host cell receptor compared to HAstV-1.

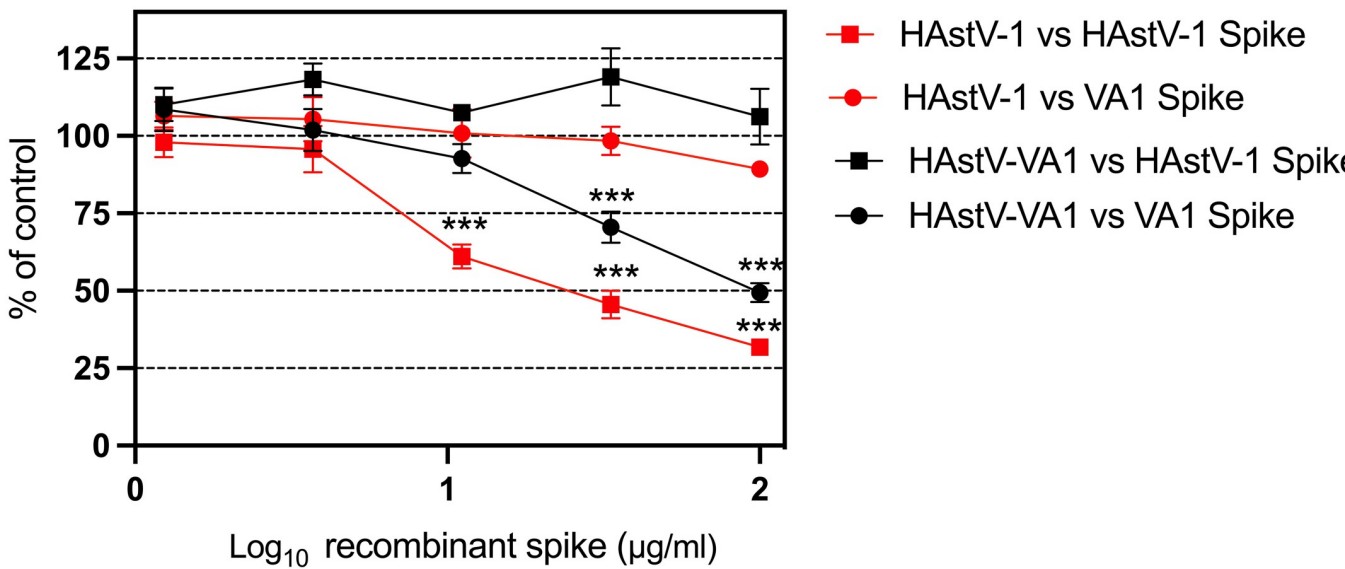

**Fig 3. *In vitro* HAstV infection competition assay.** Caco-2 cells were infected with HAstV-1 (red) or HAstV-VA1 (black) in the presence of the indicated concentration of recombinant HAstV-1 spike (squares) or VA1 spike (circles). Recombinant spike blocks infectivity by homologous HAstV but not heterologous HAstV. The data represent the HAstV infectivity in cells in the presence of each recombinant spike compared with infectivity in the absence of recombinant spike. The arithmetic means ± SEM from three independent experiments performed in duplicate are shown. ***p < 0.001.

## Mapping the cleavage site and proteins that form the mature HAstV-VA1 capsid

It was previously shown that the HAstV-VA1 capsid precursor protein VP86 is processed intracellularly into two fragments, an N-terminal fragment VP33 and a C-terminal fragment VP38 whose amino terminus was mapped to Thr348 [47]. To determine the composition of the VP33 and VP38 proteins that form the mature capsid, and to attempt to determine the C-termini of VP33 and VP38, HAstV-VA1 virus particles were purified using a CsCl gradient [42], and the capsid proteins were separated by SDS-PAGE and stained with Coomassie blue. As expected, two major bands were observed at 33 kD (VP33) and 38 kD (VP38) (Fig 4B). The corresponding bands in the gel were excised and subjected to trypsin or proalanase digestion followed by liquid chromatography and tandem mass spectrometry analysis. Shown in Fig 4A are the identified VP33 peptides colored yellow and the identified VP38 peptides colored green. It is unclear if the amino acids not observed (grey) is due to their absence in the processed proteins that compose the mature virus or to a technical challenge of identifying them by mass spectrometry; however, the expected molecular weights for the regions identified are close to the observed band sizes by SDS-PAGE (yellow VP33 amino acids would be 32.8 kD and green VP38 amino acids would be 37.1 kD).

To understand these data in the context of the mature HAstV-VA1 virion, we mapped out this sequence information onto the structures of the HAstV-VA1 capsid core and spike structural domains (Fig 4C). While the HAstV-VA1 capsid core domain structure has not been experimentally determined, it is expected to be similar to the crystal structure of the classical HAstV capsid core due to their ~40% sequence identity. Indeed, a structural alignment of an AlphaFold2 model of the HAstV-VA1 capsid core with HAstV-1 core using TM-align reveals an RMSD of 2.36 Å and TM-score of 0.91. A number of interesting observations emerged. First, VP33 amino acids encompass the N-terminal basic region and the first ~2/3 of the predicted core structural domain. The positively-charged N-terminal residues 1–71 that precede

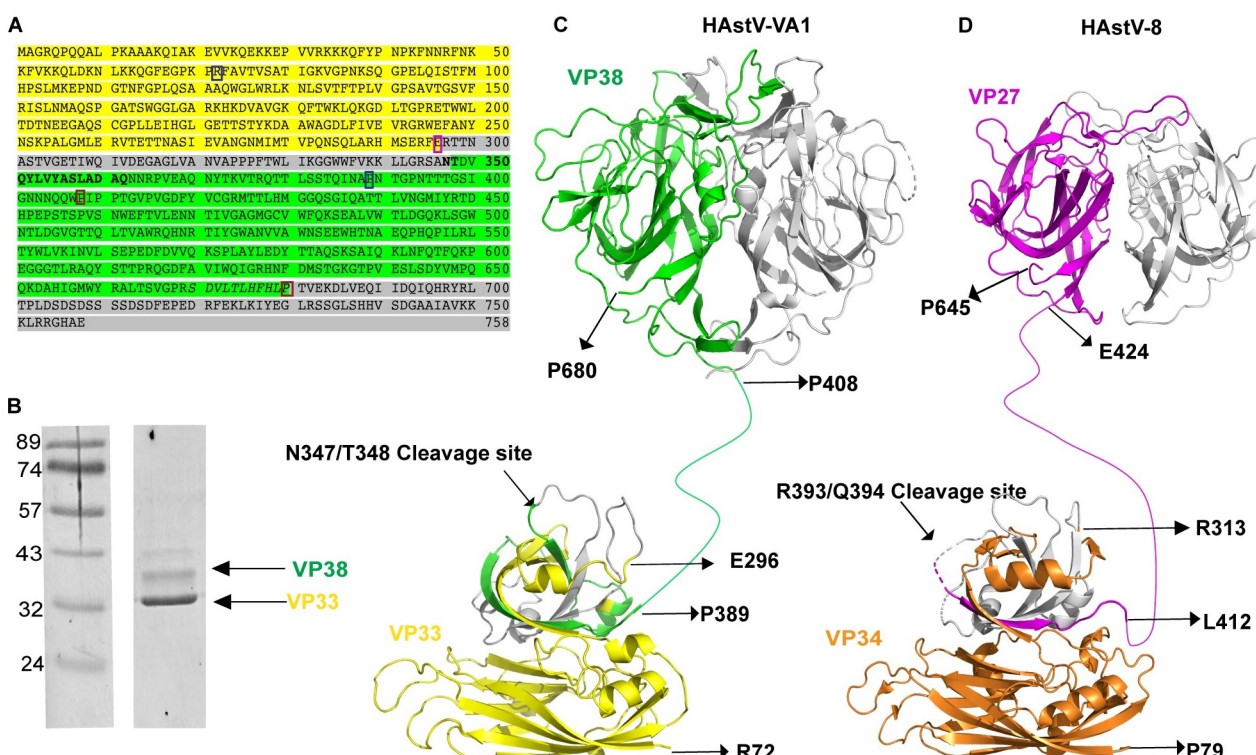

**Fig 4. Processing and assembly of the mature HAstV-VA1 capsid protein.** (A) The complete amino acid sequence of the HAstV-VA1 capsid precursor protein (GenBank accession number YP_003090288.1) is shown. Amino acids highlighted in yellow were identified by mass spectrometry as peptides of VP33, and those highlighted in green correspond to peptides of the VP38. Amino acids that are not observed are highlighted in grey. The cleavage site that generates the amino-terminal end of VP38, previously determined by Edman degradation, is shown in bold. The blue boxes correspond to the N- and C-termini of the core domain, whereas the red boxes correspond to the N- and C-termini of the spike domain. The pink box corresponds to amino acid E296, the last amino acid detected in VP33. (B) Coomassie-stained SDS-PAGE analysis of CsCl-purified virus. Lane 1: molecular weight marker, in kD. Lane 2: two bands corresponding to the HAstV-VA1 capsid proteins VP33 and VP38. Bands were excised and utilized for proteolytic digestion, liquid chromatography, and tandem mass spectrometry to identify peptides. (C) Model of the mature HAstV-VA1 capsid protein. An AlphaFold2-predicted core domain structure, a linker region, and the spike domain crystal structure are shown, colored as in panel A. The second protomer of the spike domain is colored grey. The N-terminal amino acids 1–71 that are present in VP33 are not shown. Domain termini are labeled (R72 and P389 (core domain), and P408 and P680 (spike domain)). The location of the cleavage site that results in the N-terminus of VP38 (N347/T348) is indicated. The location of E296, the last observed amino acid of VP33, which is in a structurally similar site as a known trypsin cleavage site in classical HAstVs, is indicated. (D) Model of the mature classical HAstV-8 capsid protein. The crystal structure of the HAstV-8 core domain (PDB: 5IBV), a linker region, and the crystal structure of the HAstV-8 spike domain (PDB: 3QSQ) are shown. The second protomer of the spike domain is colored grey. The N-terminal amino acids 1–76 that are present in VP34 are not shown. Domain termini are labeled (R77 and L412 (core domain), and E424 and P645 (spike domain)). The location of the trypsin cleavage site that results in the N-terminus of VP27 (R393/Q394) is indicated. The location of R313, a trypsin cleavage site, is indicated.

the core structural domain are likely inside the capsid and interact with the viral RNA genome. Residues 72–255 form the inner region of the core structural domain that has a typical jelly-roll beta-barrel fold commonly found in capsid proteins of icosahedral viruses. Residues 256–296 form a predicted beta-hairpin and an alpha-helix in the outer region of the core domain. Residues 297–347 that were not identified by mass spectrometry map to this outer region, suggesting that they are accessible on the surface of the assembled virus capsid for protease cleavage. It is interesting to note that residue E296, the last VP33 residue observed by mass spectrometry, is in a structurally similar location as a putative trypsin maturation cleavage site in classical HAstVs (R313 in HAstV-8) that is thought to be important for exposing a membrane-lytic peptide (Fig 4C and 4D) [59]. Next, we observed that VP38 amino acids include the C-terminal end of the predicted HAstV-VA1 capsid core structural domain, a predicted

linker region, and the full spike domain (Fig 4C). VP38 starts with ~42 residues that are predicted to form integral structural components of the outer region of the core domain, ensuring that the VP38 remains tethered to the icosahedral surface of the capsid. Finally, it is worth noting that the last VP38 residue observed by mass spectrometry, P680, is also the last structured amino acid in the HAstV-VA1 spike domain. Due to the size of VP38, it is predicted that the C-terminal acidic region of the HAstV-VA1 capsid is proteolytically removed intracellularly, similar to classical HAstVs that are cleaved intracellularly by caspases [47]. While there are a number of putative caspase cleavage sites (aspartates) in the HAstV-VA1 capsid C-terminus (residues 700–720), it was previously shown that the pan-caspase inhibitor Z-VAD-FMK did not affect the proteolytic processing of HAstV-VA1 capsid or production of infectious HAstV-VA1 virus [47].

## Sequence and structural comparison of HAstV-VA1[gastro] and HAstV-VA1[neuro] spikes

To date, only 9 cases of HAstV-VA1 associated encephalitis are reported, of which six of those HAstV-VA1 capsid sequences are reported [22,23,27–31,33]. To understand if sequence and/or structural changes might be driving some strains of HAstV-VA1 to be able to infect the CNS, we first evaluated the capsid spike sequence differences in five representative sequences of HAstV-VA1 associated with gastroenteritis (HAstV-VA1[gastro]) and all six of the publicly available sequences of HAstV-VA1 associated with neurological disease (HAstV-VA1[neuro]) (Fig 5A) [60,61]. We observed that the HAstV-VA1[gastro] spike sequence is generally more conserved than the HAstV-VA1[neuro] spike sequence. Specifically, only six variable sites were observed in HAstV-VA1[gastro] spike sequences, whereas 27 variable sites were observed in HAstV-VA1[neuro] spike sequences. Notably, none of these 27 residues in the HAstV-VA1[neuro] spike sequences differed in a way that might indicate that one or more of these residues was a driver for infection of the CNS. Moreover, none of these sites of variability mapped onto a specific region of the spike, rather they were scattered all over the spike (Fig 5B). Finally, to evaluate if there are structural changes in the spike that drive infection of the CNS, we aligned the HAstV-VA1[gastro] and HAstV-VA1[neuro] spike structures (Fig 5C), which resulted in an RMSD of 0.78 Å and TM-score of 0.97, revealing that no major structural differences are observed.

## Evaluation of antibody binding to HAstV-VA1[gastro] and HAstV-VA1[neuro] spikes

One reason for the higher sequence variability in the HAstV-VA1[neuro] spikes could be that the virus evolved to evade patient antibodies. A number of the patients with HAstV-VA1-associated neurological disease were reported to have received immunoglobulin therapy, which likely contains antibodies against HAstV-VA1[gastro], as there is a high seroprevalence of neutralizing antibodies to HAstV-VA1 in adults [35]. To evaluate the antigenicity of HAstV-VA1-[gastro] and HAstV-VA1[neuro] spikes, we performed a number of antibody binding assays using anti-HAstV-VA1[gastro] rabbit polyclonal serum. First, approximately equal amounts of recombinant HAstV-VA1[gastro] and HAstV-VA1[neuro] spike protein were evaluated by SDS-PAGE and Western Blot, demonstrating that the denatured spike proteins have linear epitopes for anti-HAstV-VA1[gastro] antibodies (Fig 6A). Next, a dilution-series enzyme-linked immunosorbent assay (ELISA) was performed by coating wells with recombinant HAstV-VA1[gastro] or HAstV-VA1[neuro] spike protein and evaluating binding to anti-HAstV-VA1[gastro] antibodies (Fig 6B). A difference in binding between HAstV-VA1[gastro] and HAstV-VA1[neuro] spikes was observed, although, due to limitations in sample amounts, these samples were performed only in duplicate and could not be evaluated for significance. Instead, we pursued a more

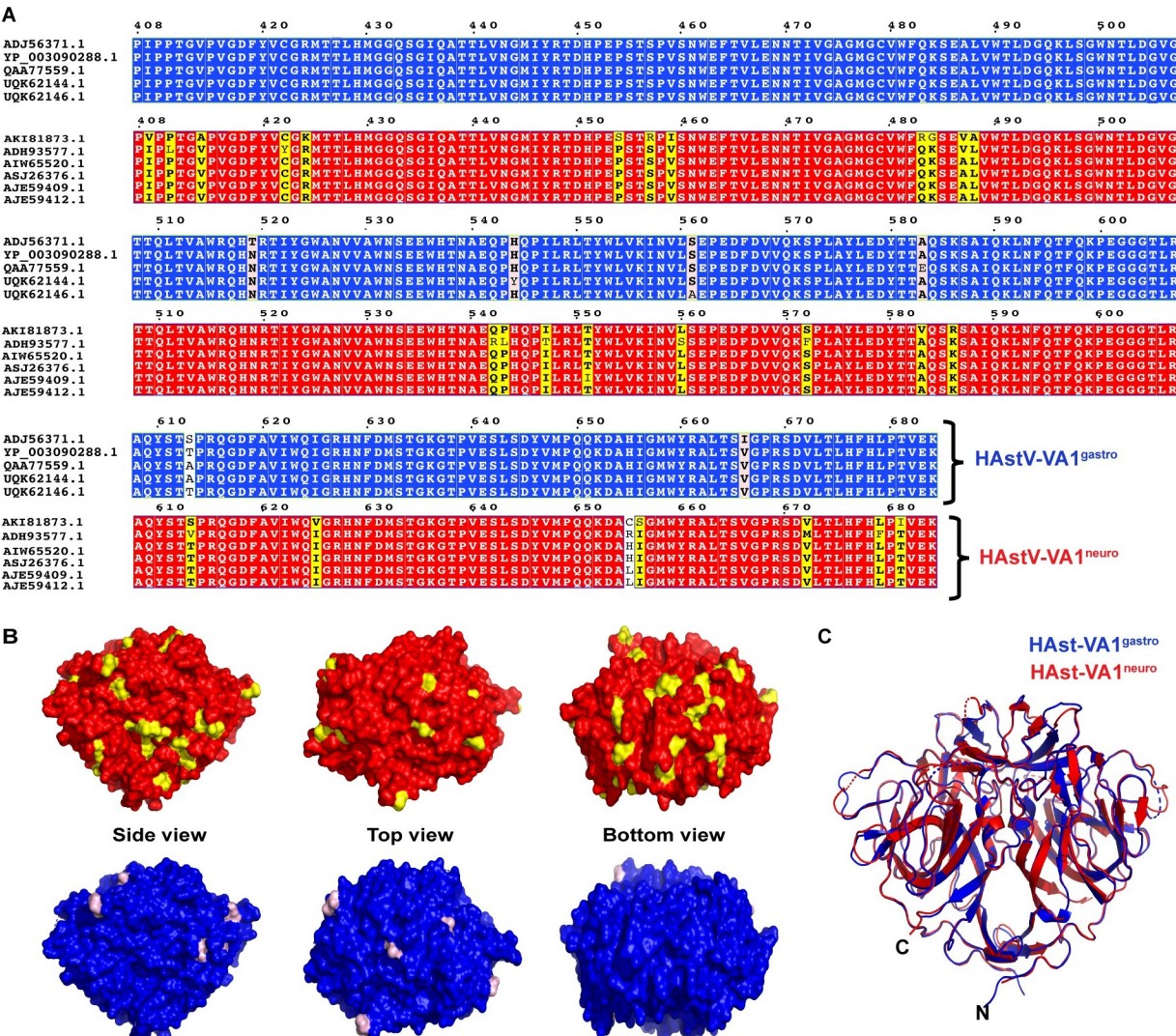

**Fig 5. Comparison of HAstV-VA1<sup>gastro</sup> and HAstV-VA1<sup>neuro</sup> spike sequences and structures.** (A) Five representative HAstV-VA1<sup>gastro</sup> spike sequences (blue) and all five available HAstV-VA1<sup>neuro</sup> spike sequences (red) aligned using the MUSCLE algorithm. Sequence differences between HAstV-VA1<sup>gastro</sup> spikes are colored light pink, and sequence differences between HAstV-VA1<sup>neuro</sup> spikes are colored yellow. (B) Location of variations (yellow and light pink) mapped onto the the VA1 spike structure, presented as a surface model from side, top, and bottom views. (C) Structural alignment between the crystal structures of the HAstV-VA1<sup>gastro</sup> spike (blue)(accession no: YP_003090288.1) and the HAstV-VA1<sup>neuro</sup> spike (red)(accession no: ADH93577.1), presented as cartoon view.

quantitative approach using a biolayer interferometry immunosorbent assay (BLI-ISA) that we have used previously to evaluate relative polyclonal antibody levels in human sera [62,63]. In the BLI-ISA, HAstV-VA1<sup>gastro</sup> and HAstV-VA1<sup>neuro</sup> spike proteins containing 10-histidine tags are loaded onto Anti-Penta-His biosensors (Fig 6C). A similar signal increase at this step confirms that equal amounts of the spike proteins are loaded onto the biosensors (Fig 6D). After a buffer step to ensure a steady baseline, the biosensors are then submerged into a 1:20 dilution of the rabbit polyclonal serum containing anti-HAstV-VA1<sup>gastro</sup> antibodies (Fig 6C). A control biosensor not coated with spike protein shows no signal at this step, demonstrating that there is no background binding of serum biomolecules to the biosensors (Fig 6C). We

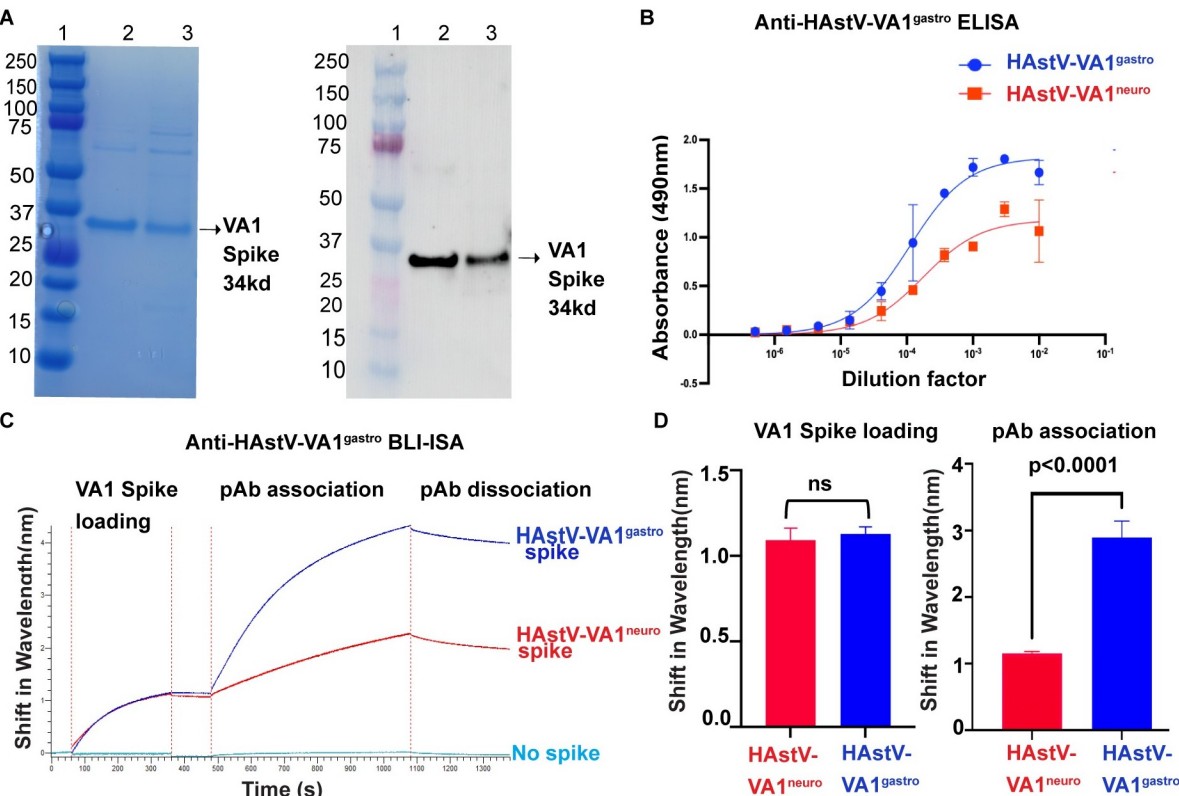

**Fig 6. Antigenic analyses of HAstV-VA1gastro and HAstV-VA1neuro spikes.** (A) Coomassie-stained SDS PAGE and anti-HAstV-VA1gastro Western Blot analyses of recombinant HAstV-VA1gastro and HAstV-VA1neuro spike proteins. Lane 1: molecular weight marker, in kD (Biorad Precision Plus Protein Dual Color Standards). Lane 2: HAstV-VA1gastro spike. Lane 3: HAstV-VA1neuro spike. (B) Anti-HAstV-VA1gastro ELISA. Wells were coated with either HAstV-VA1gastro spike (blue) or HAstV-VA1neuro spike (red), and the immunoassay was performed against a serial dilution of anti-HAstV-VA1gastro rabbit serum. Error bars indicate the standard deviation of duplicates. (C) Anti-HAstV-VA1gastro BLI-ISA. After an initial baseline step, histidine-tagged HAstV-VA1gastro spike protein (blue) or HAstV-VA1neuro spike protein (red) or no protein (cyan) were loaded onto Anti-Penta-His biosensors, followed by another baseline step. Biosensors were then dipped into a 1:20 dilution of anti-HAstV-VA1gastro rabbit serum containing polyclonal antibodies (pAb) for 10 minutes. Biosensors were then dipped into buffer to evaluate pAb dissociation. Signal changes during the HAstV-VA1 spike loading step and during the polyclonal antibody association step were measured. (D) BLI-ISA average signal changes during the HAstV-VA1 spike loading step and during the polyclonal antibody association step. Bars represent the mean of three independent experiments, and error bars indicate the standard deviation. A two-tailed T-test was performed to evaluate significance.

performed three independent BLI-ISA experiments and determined the average signal increase at the polyclonal antibody association step for each sample (Fig 6D). Statistical analyses confirm that there is a significant decrease in antibody binding to the HAstV-VA1neuro spike compared to the HAstV-VA1gastro spike (Fig 6D). Thus, these data reveal that mutations present in the HAstV-VA1neuro spike protein result in reduced anti-HAstV-VA1gastro polyclonal antibody binding.

## Discussion

In differential diagnosis of central nervous system infections, including encephalitis and meningitis, human astroviruses from the VA/HMO clade are increasingly associated with infections, particularly in immunocompromised patients, with high rates of mortality. Neurological diseases in domesticated mammals, such as in mink, ovine, bovine and porcine, are also associated with infection by VA/HMO astroviruses [13,14,17,20,64–66]. To understand how

HAstV-VAs differ structurally from classical HAstVs, we investigated the sequence-divergent HAstV-VA1 capsid spike.

We report the high-resolution structure of the HAstV-VA1 capsid spike. While the spike domain structure is similar to that of classical and MLB HAstV spikes in that it forms a homo-dimer and shares a common beta-barrel fold, the HAstV-VA1 capsid spike is larger and strikingly different in its overall shape and its biochemical surface. Notably, there are no shared conserved patches of amino acids on the surface of classical and VA1 spikes. Specifically, in classical HAstVs, the spike has an important role in virus attachment to host cells, and patches of conserved amino acids on the surface of the spike of classical HAstVs, termed P site, S site, and beta-turn, are predicted to be involved in cell attachment [52,56,57]. These conserved patches are not observed on the surface of the HAstV-VA1 spike, suggesting that HAstV-VA/HMO viruses may utilize a different receptor for cell attachment, which in turn may influence their propensity to infect CNS cells. An infection competition assay supports these structural observations since the recombinant HAstV-1 capsid spike impedes infection by HAstV-1, but not by HAstV-VA1, suggesting that these two viruses utilize a different receptor. We further investigated whether there were sequence or structural differences between the HAstV-VA1 capsid spikes from gastrointestinal and neuronal HAstV-VA1 strains that might account for the ability of the neuronal HAstV-VA1 strains to infect the CNS. However, we did not observe any conserved sequence variations within the neuronal HAstV-VA1 capsid spikes that might account for their ability to infect the CNS. Furthermore, we solved the structure of a neuronal HAstV-VA1 capsid spike and observed no major structural differences. Altogether, these studies support that HAstV-VA1 may utilize a distinct receptor for cellular infection compared to classical HAstV, and that this difference, rather than the attainment of one or more mutations, may contribute to the ability of HAstV-VA/HMO strains to more readily infect the CNS.

It is also possible that there are other differences in the HAstV-VA1 life cycle besides cell attachment that influence its ability to infect the CNS. One difference could be the capsid proteolytic processing that is required for HAstV infectivity. While classical HAstVs require extracellular trypsin for infectivity *in vitro*, HAstV-VA1 does not require extracellular trypsin for infectivity and instead becomes processed intracellularly by an unknown protease [46]. Thus, the requirement for extracellular proteases, such as those found in the intestinal tract, may limit the ability of classical HAstVs to spread beyond the intestinal tract, whereas the HAstV-VA1 does not appear to have this limitation. Despite this difference, we provide evidence showing that the proteolytic processing of the classical HAstV and HAstV-VA1 capsids appears to follow a similar path. Specifically, our mass spectrometry data support that the HAstV-VA1 capsid becomes cleaved in one or more places in the outer region of the capsid core domain, which forms the shell of the virus capsid. Recent studies suggest that this proteolytic processing in HAstVs may expose a membrane-lytic peptide utilized for host membrane disruption during cell entry [59]. Our data support that HAstV-VA1 utilizes a similar mechanism to promote infectivity. However, it is not clear how HAstV-VA1 prevents lysis of the cell upon intracellular proteolytic processing. One possibility is that proteolysis may prime the virus for entry, but another cue such as receptor engagement, endosomal acidification, proximity to membranes, and/or other factor may be required for peptide penetration of membranes. Overall, our data reveals a roadmap to understand the structural assembly of HAstV-VA1 capsid.

Beyond structural analyses, we evaluated the antigenicity of the HAstV-VA1 spike using ELISA and BLI-ISA. We had observed that neuronal HAstV-VA1 sequences had more amino acid variations than gastrointestinal HAstV-VA1 sequences, and we hypothesized that this could be due to virus evasion of human antibody responses. Our ELISA and BLI-ISA data support this hypothesis. While these results are not entirely surprising given that the antibodies

were raised against a gastrointestinal HAstV-VA1 strain, they do demonstrate that antibodies to a gastrointestinal HAstV-VA1 virus are less reactive to neuronal HAstV-VA1 spike compared to gastrointestinal HAstV-VA1 spike. Nevertheless, we cannot rule out the possibility that the observed variation in neuronal HAstV-VA1 sequences is due simply to virus intra-host variations that accumulated over the course of the infection. In fact, in most reports, the length of neurological symptoms was reported to be several weeks or months.

In a larger sense, our data provide a foundation for a number of future basic and applied studies. First, the delineation of the HAstV-VA1 capsid spike domain and methods to produce it recombinantly in bacteria open new avenues to evaluate its role in cell attachment, as was done with recombinant spikes for classical HAstV and HAstV-MLB [51,56]. Moreover, recombinant HAstV-VA1 spike can be utilized for co-precipitation studies to identify candidate host receptors. It can also be utilized as an antigen to discover monoclonal antibodies, which could be used for virus neutralization and epitope mapping studies as well as the development of antibody therapeutics for HAstV-VA1 infections. Furthermore, serological studies such as ELISAs utilizing recombinant HAstV-VA1 spike as the antigen could be used to investigate HAstV-VA1 seroprevalence in humans or evaluate different batches of intravenous immunoglobulin for therapeutic use. Finally, recombinant HAstV-VA1 spike can be evaluated as a candidate vaccine immunogen to elicit HAstV-VA neutralizing antibodies, as has been previously shown for classical HAstVs [57].

## Methods

### Phylogenetic analysis

The cladogram in Fig 1A was constructed using complete capsid protein sequences with the following NCBI accession numbers: Human astrovirus 1: AAC60723.1, Human astrovirus 2: QKW90827.1, Human astrovirus 3: QJX57344.1, Human astrovirus4: AGV40902.1, Human astrovirus 5: QKW90830.1, Human astrovirus 6: ACV92107.1, Human astrovirus 7: AAK31913.1, Human astrovirus 8: QGL54773.1, Human astrovirus MLB1: BAU68081.1, Human astrovirus MLB2: AMR45107.1, Human astrovirus MLB3: YP_006905854.1, Human astrovirus VA1 Neuronal: ADH93577.1, Human astrovirus VA1 Gastrointestinal: YP_003090288.1, Human astrovirus VA2: ACX83591.2, Human astrovirus VA3: YP_006905860.1, Human astrovirus VA4: YP_006905857.1, Human astrovirus VA5: AJI44022.1, Astrovirus VA6 UQK62274.1, Mink Astrovirus ADR65076.1, Ovine Astrovirus QDA34115.1, Bovine astrovirus CUI02224.1, Turkey astrovirus 1: AOR81715.1, Turkey astrovirus 2: NP_987088.1, Turkey astrovirus 3: AAV37187.1.

### HAstV-VA1 capsid spike multiple sequence alignments

The multiple sequence alignment in Fig 5A was prepared using five representative gastrointestinal HAstV-VA1 capsid spike sequences with the following NCBI accession numbers: Human Astrovirus VA1 ADJ56371.1, Human Astrovirus YP_003090288.1, Human Astrovirus VA1 QAA77559.1, Human Astrovirus VA1 UQK62144.1, and Human Astrovirus VA1 UQK62146.1, and all 6 available neuronal HAstV-VA1 capsid spike sequences with the following NCBI accession numbers: Human Astrovirus VA1 AKI81873.1, Human Astrovirus VA1 ADH93577.1, Human Astrovirus VA1 AIW65520.1, Human Astrovirus VA1 ASJ26376.1, Human Astrovirus VA1 AJE59409.1, and Human Astrovirus VA1 AJE59412.1 using ESPript 3.0 [61] which is an online server and renders sequence similarities using pre-aligned sequences. The pre-aligned spike sequences were generated in AliView software using MUSCLE algorithm [60].

## Expression, purification, limited proteolysis, and mass spectrometry of neuronal HAstV-VA1 capsid C-term

A codon-optimized synthetic gene encoding the neuronal HAstV-VA1 capsid residues 394–758 (NCBI accession number ADH93577.1) termed HAstV-VA1 capsid C-term was cloned into the plasmid pBacPAK8 in frame with an N-terminal 10-histidine tag. Recombinant baculovirus stocks were generated using the flashBAC system (Mirus Bio). Sf9 insect cells in ESF921 media (Expression Systems) at a density of 2 million viable cells/mL were infected with 0.025 mL of baculovirus stock/mL culture and cultured at 180 rpm at 27°C for 4 days. Cells were harvested by centrifugation, resuspended in buffer A (10 mM Tris-HCl pH 8.0, 300 mM NaCl, 20 mM imidazole, 2 mM MgCl2, EDTA-free protease inhibitor cocktail (Millipore), and 0.0125 U/ml benzonase (Millipore)), and lysed by sonication. The lysate was clarified by centrifugation (40,000 $g$), 0.22μm-filtered, and the HAstV-VA1 capsid C-term protein was purified from the supernatant using TALON metal affinity chromatography. Limited proteolysis with trypsin protease was used to identify a trypsin-stable fragment. Briefly, HAstV-VA1 capsid C-term protein (~43kD) was incubated with 0.4 w/w trypsin overnight at 4°C. A ~36kD fragment was observed by Coomassie-stained SDS-PAGE. The trypsin-digested protein sample (20 μl) was injected at flow rate of 200 μl/min onto an HPLC with a reverse-phase column, 100mm x 2.1mm id, Proto 300 C4 (Higgins Analytical, Inc.) with a 5μm particle size. The mobile phase consisted of solvent A (0.1% formic acid in HPLC grade water) with a gradient to solvent B (0.1% formic acid in acetonitrile). The samples were analyzed by using a linear ion trap mass spectrometer system (LTQ, Thermo Finngan). Protein and peptides were detected by full scan MS mode (over the m/z 300–2000) in positive mode. The electrospray voltage was set to 5 kV. Deconvoluted ESI mass spectra of the reversed-phase peaks were generated by Magtran software. The mass of the fragment, 36,020 Daltons, was very close to a theoretical tryptic fragment of 35,970 Daltons generated by cleavage after arginine 697.

## Expression and purification of neuronal HAstV-VA1 spike

A synthetic gene encoding the neuronal HAstV-VA1 capsid residues 394–697 was cloned into pBacPAK8 in frame with an N-terminal 10-histidine tag, superfolder GFP, and a thrombin protease cleavage site (termed sf-GFP HAstV-VA1 spike). Recombinant sf-GFP HAstV-VA1 spike was expressed in Sf9 insect cells and purified the same as was done for HAstV-VA1 capsid C-term. Recombinant sfGFP-VA1 spike was digested overnight with thrombin to remove the sf-GFP and dialyzed into 10 mM Tris-HCl pH 8.0 and 150 mM NaCl. The neuronal HAstV-VA1 capsid spike (HAstV-VA1[neuro] spike) was further purified by size exclusion chromatography on a Superdex 200 16/600 column (Cytiva). Fractions containing the HAstV-VA1[neuro] spike were concentrated to 2.3 mg/ml.

## Expression and purification of gastrointestinal HAstV-VA1 spike

A synthetic gene encoding the gastrointestinal HAstV-VA1 capsid spike (HAstV-VA1[gastro] spike) residues 408–684 (NCBI accession number YP_003090288.1) was cloned into pET52b in frame with a C-terminal thrombin protease cleavage site and 10-histidine tag. The plasmid was transformed and grown in *E. coli* strain T7 Express (New England Biolabs) to an optical density reached 0.6, and protein production was induced with 1 mM isopropyl-D-thiogalacto-pyranoside (IPTG) at 18°C for 18 h. Cells were harvested by centrifugation and lysed by sonication in Buffer A (20 mM Tris-HCl pH 8.0, 500 mM NaCl, 20 mM imidazole) containing 2 mM MgCl$_2$, 0.0125 U/ml benzonase (Millipore), and EDTA-free protease inhibitor cocktail (Roche). The cell lysate was clarified by centrifugation (40,000 $g$), 0.22μm-filtered, and the

HAstV-VA1$^{gastro}$ spike was purified from the supernatant using TALON metal affinity chromatography. The protein was digested overnight with thrombin to remove the 10X-histidine tag and dialyzed into 20 mM Tris-HCl pH 8.0 and 150 mM NaCl. The HAstV-VA1$^{gastro}$ spike protein was purified by size exclusion chromatography on a Superdex 200 increase 10/300 column (Cytiva) and concentrated to 6.0 mg/mL. The oligomeric state of the recombinant HAstV-VA1$^{gastro}$ spike was estimated by comparing its retention volume to those of Gel Filtration Standards (Bio-Rad) on the Superdex 200 increase 10/300 column.

## Structure determination of gastrointestinal and neuronal HAstV-VA1 spike proteins

The HAstV-VA1$^{neuro}$ spike protein was crystallized in 0.1 M HEPES pH 7.5 and 22.5% PEG3350 using the hanging drop method, and cryoprotected using 0.1 M HEPES pH 7.5, 27.5% PEG3350, and 25% glycerol. The HAstV-VA1$^{gastro}$ spike was crystallized in 0.2 M MgCl$_2$, 0.1 M Tris-HCl pH 8.5, and 25% PEG3350 using the hanging drop method, and cryoprotected using 0.2 M MgCl$_2$, 0.1 M Tris-HCl pH 8.5, 25% PEG3350, and 25% Ethylene glycol. All crystals were flash-frozen in liquid nitrogen and diffraction data from a single crystal were collected at cryogenic temperature using a wavelength of 0.97 Å at the Advanced Light Source beamline 5.0.1 for the HAstV-VA1$^{neuro}$ spike and a wavelength of 1.03 Å at the Advanced Photon Source beamline 23ID-D for the HAstV-VA1$^{gastro}$ spike. The data were processed with XDS for the HAstV-VA1$^{neuro}$ spike or Mosflm for the HAstV-VA1$^{gastro}$ spike and scaled with Aimless [67–69]. CC$_{1/2}$ and I/σI statistics were used to select the 2.73 Å resolution cutoff for the HAstV-VA1$^{neuro}$ spike data. The structure of the HAstV-VA1$^{neuro}$ spike protein was solved by molecular replacement in Phenix using part of a model generated by AlphaFold2 for the 2022 CASP15 competition. While the full AlphaFold2 model of the HAstV-VA1$^{neuro}$ spike did not yield a molecular replacement solution, deletion of amino acids with pLDDT confidence scores less than 55 (58 out of 272 amino acids) yielded a model that gave a partial molecular replacement solution (LLG = 137). The final HAstV-VA1$^{neuro}$ spike structure was refined and manually built using Phenix and Coot [69–71]. The structure of the HAstV-VA1$^{gastro}$ spike was solved by molecular replacement in Phenix using the crystal structure of the HAstV-VA1$^{neuro}$ spike protein as a starting model. The final HAstV-VA1$^{gastro}$ spike structure was refined and manually built using Phenix and Coot [69–71].

## HAstV infection competition assay with recombinant HAstV spike proteins

A HAstV infection competition assay was utilized to evaluate infection inhibition by recombinant spike proteins [51]. Briefly, Caco-2 cells were grown in 96-well tissue culture treated plates until confluence. The growth medium was removed and replaced by MEM pre-cooled to 4°C, and the cells were incubated for 20 min on ice. MEM was then replaced by virus diluted in MEM at an MOI of 0.02 and incubated on ice for 1 h. Unbound virus was washed away with MEM and the indicated concentration of the purified HAstV spike proteins, diluted in MEM, was added and incubated for 1 h on ice. The plates were then transferred to 37°C for 1 h, washed with MEM to remove the proteins, and then were incubated in DMEM supplemented with non-essential amino acids for 18 h for HAstV-1-RIVMb [72] or for 24h for HAstV-VA1. After this time, the cells were processed by an immunoperoxidase assay to detect the infected cells, as previously described [72] with some modifications. Briefly, the cells were fixed at room temperature for 20 min with 2% formaldehyde diluted in PBS and permeabilized with 0.2% Triton X-100-PBS for 15 min. To stain the cells infected with HAstV-1, a rabbit polyclonal serum to HAstV-1 was used and for cells infected with HAstV-VA1 we employed a

rabbit polyclonal serum raised against HAstV-VA1 [72]. Three independent experiments were performed in duplicate.

## ELISA

ELISA plates (Corning 3590) were coated over two days at 4˚C with 50 μL per well of 2 μg/mL HAstV-VA1$^{gastro}$ spike in PBS, 2 μg/mL HAstV-VA1$^{neuro}$ spike in PBS, or 2 μg/mL control antigen BSA in PBS. After the two-day incubation, the antigen solutions were removed by aspiration and wells were washed three times with PBS-T (PBS with 0.1% Tween 20). The plates were blocked for one hour at room temperature with 200 μL per well of blocking buffer (5% non-fat milk in PBS-T). After removing the blocking buffer, primary polyclonal antibody in hyperimmune rabbit serum to HAstV-VA1$^{gastro}$ virus, first diluted 1:100 in blocking buffer and then diluted 1:3 in series in blocking buffer, was added to wells (140 μl/well) and incubated for two hours at room temperature. After incubation, the primary antibody was removed by aspiration and wells were washed three times with PBS-T. Next, a 1:3000 dilution of secondary antibody (anti-rabbit-IgG conjugated to horseradish peroxidase) (Thermo Fisher Scientific 31462) was prepared in 1% non-fat milk in PBS-T, and 50 μl of this secondary antibody was added to each well and incubated at room temperature for 1 hour. The plate was again washed three times with PBS-T and 100 μl of an OPD (o-phenylenediamine dihydrochloride) solution was added to each well. This substrate was left on the plate for 8 minutes and then the reaction was stopped by the addition of 50 μl per well of 3 M hydrochloric acid. The optical density at 490 nm (OD490) was measured using a Molecular Devices SPECTRAmax PLUS 384 plate reader. Background values (~0.05) from antigen-coated wells lacking primary antibody were subtracted from all data prior to curve fitting in Prism. Samples were performed in biological duplicates and error bars represent one standard deviation from the mean.

## Biolayer interferometry binding experiments

An Octet RED384 was used for collection of Biolayer interferometry data using the Data Acquisition Software (version 11.1.1.19). All Binding experiments were performed in assay buffer (PBS, 1% BSA, 0.05% Tween 20). Purified HAstV-VA1$^{gastro}$ and HAstV-VA1$^{neuro}$ spike proteins were diluted to a concentration of 2 μg/mL in assay buffer. Hyperimmune rabbit serum to HAstV-VA1$^{gastro}$ virus was diluted 1:20 in assay buffer. Binding assays were performed at room temperature with the plate shaking at 1000 rpm. Anti-Penta-His biosensors were pre-equilibrated for 10 min in assay buffer. To run the assay, biosensors were dipped in assay buffer for 1 min to obtain a baseline and then dipped in HAstV-VA1 spike proteins for 5 min to load the histidine-tagged antigens onto the biosensors. Next, biosensors were dipped in assay buffer for 2 min to obtain a baseline and then dipped into the 1:20 diluted rabbit sera for 10 min to measure association of serum antibodies. Dissociation was evaluated by dipping the biosensors into assay buffer for 5 min. To quantify serum antibody binding, the total signal increase during the association step was determined. Samples were performed in biological quadruplicates and error bars represent one standard deviation from the mean.

## LC-MS/MS of HAstV-VA1$^{gastro}$ capsid proteins

HAstV-VA1$^{gastro}$ virus was grown in Caco-2 cells as described previously [47]. Viral particles purified by CsCl isopycnic centrifugation were analyzed by 11% SDS-PAGE. The gel was stained with Coomassie brilliant blue R-250 (Sigma) and imaged with a laser scanner (Typhoon FLA 9500, GE Healthcare) using near-infrared emission. The VP33 and VP38 protein bands were cut out, and the polyacrylamide slices were sent to the Proteomics Facility at the Montreal Clinical Research Institute (Canada). The samples were prepared, digested with

either trypsin or ProAlanase, and analyzed by nanoscale liquid chromatography coupled to tandem mass spectrometry (nano-LC-MS/MS).

## Supporting information

**S1 Fig. Electron density maps of the HAstV-VA1$^{gastro}$ spike and the HAstV-VA1$^{neuro}$ spike.** Electron density maps (slate blue) are contoured at 1.0σ around the indicated amino acids. Regions were selected to highlight the electron density around amino acids that differ between the strains, for example at (A) amino acids 421–423, (B) amino acids 487–489, and (C) amino acids 572–574.
(PDF)

## Acknowledgments

We thank Dr. John Dzimianski and Dr. Sarvind Tripathi for assistance on X-ray diffraction data collection and structure determination. We thank Qiangli Zhang for assistance with mass spectrometry experiments at the UCSC Mass Spectrometry Facility to analyze the recombinant HAstV-VA1 capsid limited proteolysis samples. We thank Rafaela Espinosa for preparing the anti-HAstV-VA1 rabbit polyclonal serum. Beamline 5.0.1 of the Advanced Light Source, a DOE Office of Science User Facility under Contract No. DE-AC02-05CH11231, is supported in part by the ALS-ENABLE program funded by the National Institutes of Health, National Institute of General Medical Sciences, grant P30 GM124169-01. This research used resources of the Advanced Photon Source, a U.S. Department of Energy (DOE) Office of Science User Facility operated for the DOE Office of Science by Argonne National Laboratory under Contract No. DE-AC02-06CH11357. Funding for the purchase of the Octet RED384 instrument was supported by the NIH S10 shared instrumentation grant 1S10OD027012-01. Funding support for the UCSC Mass Spectrometry Facility was provided by the Thermo Electron Corporation (seed funds), the W.M. Keck Foundation (grant 001768), and NIH's National Center for Research Resources (grant S10RR020939).

## Author Contributions

**Conceptualization:** Carlos F. Arias, Rebecca M. DuBois.

**Funding acquisition:** Carlos F. Arias, Rebecca M. DuBois.

**Investigation:** Anisa Ghosh, Kevin Delgado-Cunningham, Tomás López, Kassidy Green.

**Supervision:** Carlos F. Arias, Rebecca M. DuBois.

**Visualization:** Anisa Ghosh.

**Writing – original draft:** Anisa Ghosh, Rebecca M. DuBois.

**Writing – review & editing:** Anisa Ghosh, Kevin Delgado-Cunningham, Tomás López, Kassidy Green, Carlos F. Arias, Rebecca M. DuBois.

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
