## [Decision Letter · Decision Letter 0]

16 Nov 2023

Dear Dr. DuBois,

Thank you very much for submitting your manuscript "Structure and antigenicity of the divergent human astrovirus VA1 capsid spike" for consideration at PLOS Pathogens. As with all papers reviewed by the journal, your manuscript was reviewed by members of the editorial board and by several independent reviewers. In light of the reviews (below this email), we would like to invite the resubmission of a significantly-revised version that takes into account the reviewers' comments.

We cannot make any decision about publication until we have seen the revised manuscript and your response to the reviewers' comments. Your revised manuscript is also likely to be sent to reviewers for further evaluation.

Sincerely,

Félix A. Rey

Academic Editor

PLOS Pathogens

Guangxiang Luo

Section Editor

PLOS Pathogens

Kasturi Haldar

Editor-in-Chief

PLOS Pathogens

orcid.org/0000-0001-5065-158X

Michael Malim

Editor-in-Chief

PLOS Pathogens

orcid.org/0000-0002-7699-2064

Editorial Review:

In this manuscript, Rebecca DuBois and colleagues describe the structure of the spike domain of a highly divergent clade of human astroviruses, termed VA/VMO. All human astroviruses are associated with gastroenteritis in children, but the VA/VMO clade can also cause severe neurological symptoms in immuno-compromised patients. Although there have been structural studies on the classical astroviruses, the VA/VMO strain has not been studied so far. The authors show that the core domain of the capsid shares 40% sequence identity with the classical clade, but the spike domain displays no sequence similarity. They therefore conducted a systematic study to identify the region of the protein corresponding to the capsid domain and determine its X-ray structure. They show that although the overall fold is the same as that of its counterpart in the classical astroviruses, there are many more loops at the surface, and it has a bigger size. Moreover, none of the conserved patches associated with interaction with host entry factors for entry are absent, suggesting that the VA/VMO astroviruses may use different factors to gain entry into cells. The authors also identified the activating cleavage sites of the capsid protein required to become competent for entry. The identification of the domain boundaries allowed the authors to also produce and crystallize the spike domain of a VA/VMO strain not associated with neurological symptoms, showing no difference in structure.

The reported study provides very important insight into the organization of this particular clade of astroviruses. The authors observe that examination of all the available sequences of strains associated with neurological symptoms display more variation at the surface than those isolated from patients with gastrointestinal symptoms. These mutations appear randomly scattered at the surface, suggesting that the immunocompromised individuals, not being able to control replication, allow the virus to replicate unchecked and accumulate mutations (as astroviruses are RNA viruses, their polymerase has no proofreading ability). Furthermore, immunocompromised patients are often treated with antibodies that select for surface variants. The authors indeed show by BLI that a polyclonal rabbit antibody binds less well to the “neuro” strain than to the “gastro” strain. Yet, since in the Methods section they say that they used “hyperimmune rabbit serum to HAstV-VA1gastro virus”, this result was to be expected: the differences all map to the exposed spike surface, and so the serum will bind to the particles that were used to elicit it better than those that have multiple mutations at their surface.

Overall, what appears to emerge from the study is that viruses in this clade are potentially likely to use different receptors than classical astroviruses, and that left unchecked in immunocompromised individuals, they will cause sufficiently high viremia to reach the central nervous system. Yet this conclusion is not clearly stated, as the authors simply say that their study “provides insights into neurotropism and immune escape” without explaining what this insight is. This needs to be fixed, both in the abstract and at the end of the introduction (where they state it twice, in a redundant way).

In conclusion, the studies on this important human pathogen reported in this manuscript are significant as they provide important new clues about the organization of this distant astrovirus clade for which almost nothing was understood, beyond knowing the amino acid sequences.

I have, however, issues with the current presentation, described below:

1- The abstract and the Author’s summary say the same thing. The latter could be used to place the context of the study, rather than repeating what is described in the abstract.

2- The conclusion of the study should be spelt out in the abstract, beyond saying that it provides new insight without stating which insights.

3- The identification of the right boundaries for the spike domain should allow the authors to make a construct that diffracts better than the one for the “neuro” strain, for which the crystallographic data are sub-optimal.

4- Since an important result of the study is the characterization of the way the VP86 precursor protein is characterized, with the boundaries of the mature VP33 and VP38 proteins, the authors should present side-to-side their results with those obtained earlier for the classical astroviruses (which have a VP90 precursor that leads to mature VP34 and VP27 proteins). A structural alignment of the corresponding proteins between the classical and the VO/VMA clade, highlighting the common secondary structures and maturation cleavage sites would be very useful.

Finally, here I pick some sentences of the manuscript that should be fixed:

In the abstract (line 28) and Discussion (line 331) the authors write “suggesting that HAstV28 VA1 utilizes a different cell receptor.” I recommend writing “suggesting that it may utilize a different receptor”, to make clear that this is not known.

Lines 40-43: “The human astrovirus VA1 spike structure reveals major differences in size, shape, and structural features compared to the classical human astrovirus spike, suggesting that the divergent strains have evolved different mechanisms to attach to host cells”. This sentence needs rephrasing. What is meant by “major” here? Certainly not a different fold, as the secondary structure is the same. And does it necessarily mean that the attachment mechanism will be different? As an exemple that contradicts their assertion, the human alphacoronavirus N63 has a receptor binding domain in the spike protein that is unrelated to that of SRAS0CoV-2, yet both use human ACE2 as receptor. So, I urge the auhtors to be more cautious in their assertions.

Line 120: “These structures provide insights into HAstV121 VA entry, evolution, maturation, tropism, and antigenicity.” As indicated above, please explain what these insights are, rather than repeating that there are new insights,

Lines 127-129: “Mapping of variants within the spike domain of gastrointestinal and neuronal HAstV-VA1 strains yields insights into virus tropism and antigenicity, which are further supported by antigenic studies.” Same as above

129-132: “Overall, these studies provide valuable new insights that lay a foundation for the understanding of HAstV-VA biology and the development of vaccines and therapeutics against this divergent HAstV.” Same as above.

Lines 203-206: the loops and ying/yang motifs are not very apparent in the Figure. Please make an improved Figure, better highlighting the motifs mentioned in the text.

Reviewer's Responses to Questions

**Part I - Summary**

Reviewer #1: The manuscript by Gosh, et al., describes the structure of the VA1 astrovirus capsid spike using crystallography and sequence alignments. In immunocompromised patients, the virus can infect the nervous system. Therefore, the goal here was to compare those strains that cause limited gastrointestinal symptoms with those that cause neuropathology.

In brief, there are some interesting results presented here, but some major questions remain with regard to the quality of the neuro structure and impact of the results. Therefore, regrettably, I have to recommend ‘reject’.

Problems that need to be addressed.

1) One problem of concern was some of the data statistics shown in Table 1. The Rmerge statistics for the neuro strain are horrible. As written in the table, the overall Rmerge for the data is 35%, which is already very high, and the Rmerge for the highest resolution shell (2.7Å) is 210%. I frankly have never seen published numbers this bad. This not only clearly says that their resolution is not 2.7Å but, to me, casts doubt on their structure and its interpretation. Not surprisingly, the overall B factor for this structure is about 4 times that of the gastro strain. Solutions: A) the authors need to cite a realistic resolution that may impact how much it can be used in comparative analyses. B) example electron density needs to be shown so that the reader can evaluate the cited differences.

2) If using the region of 408-684 of gastro crystallized into such highly diffracting crystals, why was this not tried with the neuro? Clearly, there are issues with the neuro crystals. If this was attempted and failed, that should be noted. Therein could lie a major difference between neuro and gastro. What if there is not a structural difference between neuro and gastro but a stability difference not observable in a crystal structure?

3) Gastro has six variable AA sites compared to other astroviruses whereas neuro has 27. These differences between neuro and gastro are scattered all over the spike. While the authors conclude that one or more of these differences are correlated with tissue tropism, there is not data to support this. This comes back to a nagging question I had with this study. Neuropathology is only seen in immunocompromised patients. I would posit that it is possible that both strains are able to infect the nervous system but is neutralized in normal patients before it can spread to that tissue. Therefore, are we really looking at fortuitous tropism differences or true differences in the capsids? Indeed, the antibody binding analyses suggest that the major differences between neuro and gastro are likely due to divergence due to immunotherapy in the immunocompromised neuro patients. This is a major concern to me since it speaks to impact of these studies.

Reviewer #2: In this manuscript, Ghosh et al., describe the structures of the spike domain of the human astrovirus (HAstV) belonging to the VA/HMO clade that had remained elusive. HAstvs, classified into three clades (classical, MHV, VA/HMO), generally cause gastroenteritis in children and more severe infections and symptoms in immunocompromised patients. However, some strains of astroviruses belonging to the VA/HMO clade are also associated with neurological diseases such as meningitis and encephalitis. In this study, authors, by systematically analyzing the spike sequences of VA1 strains followed by limited trypsin digestion, after several attempts at expression and purification, succeeded in determining the crystallographic structures of the spike domain of HAstv isolated from a gastrointestinal infection and from a patient with neurological disease. While the spike structure of the gastro strain was determined at 1.46 A, that of the neuro strain was determined at 2.73 A . Both structures are dimeric, as expected, and despite other expected similarities with the previously determined spike structures of classical and MHV clades, they show marked differences in both size and overall shape. These differences suggest that HAstV-VA1 likely engages a different receptor for cell attachment, which could impact tropism. In addition, using N-terminal sequencing and mass spectrometry analyses of mature HAstV-VA1, authors identify the cleavage site that indicates domain organization in the mature virion and how it compares with classical strains. Authors show that although gastro and neuro HAstV-VA1 spike structures are very similar, the neuro sequences have more amino acid variations compared to gastro strains, and using ELISA and BLI-ISA they demonstrate that antibodies to a gastrointestinal HAstV-VA1 virus are less reactive to neuronal HAstV-VA1 spike compared to gastrointestinal HAstV-VA1 spike.

In summary, the study provides novel insights into how the spike structures of VA/HMO clade differ from that of classical HAstV and how, despite similar structures, the neuro and gastro HAstV might vary in their antigenic response. This study lays a strong foundation for further in-depth studies for receptor identification and the discovery of neutralizing antibodies and epitope mapping.

The manuscript is well-written, with a good background, adequate methodological details, well-described results, an appropriate discussion, and well-designed figures. The results are convincing, although their implications for why certain strains are associated with neurological diseases are tenuous.

Other comments:

Abstract: lines: 29-33. These sentences, in present continuous tense (also lines 129-130), are a bit odd to read and non-informative. Authors should clearly state what they learned from these studies.

Table 1: Overall Rmerge of ~35% and 210% for the highest resolution shell for the neuro data indicate poor quality and likely much lower resolution. The Rwork/Rfree values and their spread also are a bit problematic. Authors should include some explanation and comments if this affects the modeling and comparison with gastro structure.. .

Given high multiplicity, Rpim values for both the data sets should be included in the table.

Lines 179, 366, 367, 456: gastrointestinal is pelt wrong.

**Part II – Major Issues: Key Experiments Required for Acceptance**

Reviewer #1: 1) There are issues with the structure of the neuro strain that need to be addressed.

2) While probably beyond the scope of these studies, they need to demonstrate that there are truly differences in tissue tropism - probably via tissue culture analyses in the absence of an immune response. This speaks to the impact of these studies, which is a major concern.

Reviewer #2: None

**Part III – Minor Issues: Editorial and Data Presentation Modifications**

Reviewer #1: (No Response)

Reviewer #2: As suggested in my comments above

PLOS authors have the option to publish the peer review history of their article (what does this mean?). If published, this will include your full peer review and any attached files.

Reviewer #1: No

Reviewer #2: No
---

## [Editor Report · Decision Letter 1]

29 Jan 2024

Dear Dr. DuBois,

Thank you very much for submitting your manuscript "Structure and antigenicity of the divergent human astrovirus VA1 capsid spike" for consideration at PLOS Pathogens. The editorial board appreciated the revisions and we are happy to say that we are likely to accept this manuscript for publication, providing that you modify it following the editorial review below.

Editorial review

The authors have adequately addressed most of the issues raised by the reviewers of the original submission. They have incorporated additional data that strongly supports their hypothesis that HAstV-VA1 may use a different host receptor (Figure 3). They have also added an additional panel to compare the organization and location of maturation cleavages found for HAstV-VA1 to the classical HAstV. In addition, they provide a supplementary Figure with instances of the electron density for the structure that one of the reviewers had call into question. They have also clarified a number of statements that were not clear in the initial version, and as a consequence, the revised version is much improved and the impact of their results is more evident.

I only have very minor comments that the authors can easily address in a final version.

The one remaining issue has to do with the results of the BLI-ISA with rabbit hyperimmune serum. Although they acknowledge in the response to the reviewer that this difference was expected as this serum was raised against the "gastro" variants, the authors omitted to state this in the text. This needs to be fixed.

Also, in the discussion on the role of trypsin cleavage to release a membrane-lytic peptide that is important for virus entry (paragraph around line 380), something is still not obvious: if HAstV-VA1 undergoes this cleavage intracellularly in the producer cell, the released peptide would presumably lyse the producer cell, but will not be available when it reaches the target cell to be used for entry. It is important to address this issue in the Discussion.

In the Methods section, the statement saying that the multiple amino acid sequence alignment is in Figure 4A needs to be updated, as in the revised version it is Figure 5A (line 428). Also, reading the corresponding paragraph in the Methods, it appears that “all the available neuronal VA1 capsid sequences” amount to six sequences, which is comparable with the five "gastro" sequences used. It is important to state the number of sequences used in the main text (around line 287), as “all the available sequences" is very vague; if there were hundreds of sequences available, it would be normal to find many more differences than in the comparison of 5 sequences of the "gastro" strains.

Figure 2: please add rotation symbols to indicate the transformations relating bottom and middle panels.

Line 107: “~35nm T=3 icosahedral capsid” This is the particle diameter, I presume. Please specify.

Lines 116-117: “Notably, while the classical HAstV-1, -2, and -8 spike structures are structurally homologous (RMSD ~1.2 Å)” The word “homologous” (which means that the spike proteins derive from a common ancestor) is misused here, what the authors mean is that the structure of the MLB1 spike has diverged further, but it is still bears homology to the others as the overall fold is retained. Please correct.

Line 213: “sequence homology” should be “sequence similarity”. I urge the authors to read the following commentary on the misuse of the term homology in the literature (https://doi.org/10.1016/0092-8674(87)90322-9)

Line 216: that classical and HAstV-VA1 “MAY” utilize a different host cell receptor.

Sincerely,

Félix A. Rey

Academic Editor

PLOS Pathogens

Guangxiang Luo

Section Editor

PLOS Pathogens

Michael Malim

Editor-in-Chief

PLOS Pathogens

orcid.org/0000-0002-7699-2064

The authors have adequately addressed most of the issues raised by the reviewers of the original submission. They have incorporated additional data that strongly supports their hypothesis that HAstV-VA1 may use a different host receptor (Figure 3). They have also added an additional panel to compare the organization and location of maturation cleavages found for HAstV-VA1 to the classical HAstV. In addition, they provide a supplementary Figure with instances of the electron density for the structure that one of the reviewers had call into question. They have also clarified a number of statements that were not clear in the initial version, and as a consequence, the revised version is much improved and the impact of their results is more evident.

I only have very minor comments that the authors can easily address in a final version.

The one remaining issue has to do with the results of the BLI-ISA with rabbit hyperimmune serum. Although they acknowledge in the response to the reviewer that this difference was expected as this serum was raised against the "gastro" variants, the auhtors omitted to state this in the text. This needs to be fixed.

Also, in the discussion on the role of trypsin cleavage to release a membrane-lytic peptide that is important for virus entry (paragraph around line 380), something is still not obvious: if HAstV-VA1 undergoes this cleavage intracellularly in the producer cell, the released peptide would presumably lyse the producer cell, but will not be available when it reaches the target cell to be used for entry. It is important to address this issue in the Discussion.

In the Methods section, the statement saying that the multiple amino acid sequence alignment is in Figure 4A needs to be updated, as in the revised version it is Figure 5A (line 428). Also, reading the corresponding paragraph in the Methods, it appears that “all the available neuronal VA1 capsid sequences” amount to six sequences, which is comparable with the five "gastro" sequences used. It is important to state the number of sequences used in the main text (around line 287), as “all the available sequences" is very vague; if there were hundreds of sequences available, it would be normal to find many more differences than in the comparison of 5 sequences of the "gastro" strains.

Figure 2: please add rotation symbols to indicate the transformations relating bottom and middle panels.

Line 107: “~35nm T=3 icosahedral capsid” This is the particle diameter, I presume. Please specify.

Lines 116-117: “Notably, while the classical HAstV-1, -2, and -8 spike structures are structurally homologous (RMSD ~1.2 Å)” The word “homologous” (which means that the spike proteins derive from a common ancestor) is misused here, what the authors mean is that the structure of the MLB1 spike has diverged further, but it is still bears homology to the others as the overall fold is retained. Please correct.

Line 213: “sequence homology” should be “sequence similarity”. I urge the authors to read the following commentary on the misuse of the term homology in the literature (https://doi.org/10.1016/0092-8674(87)90322-9)

Line 216: that classical and HAstV-VA1 “MAY” utilize a different host cell receptor.

Reviewer Comments (if any, and for reference):

Figure Files:

Data Requirements:

Reproducibility:

References:

---

## [Editor Report · Decision Letter 2]

5 Feb 2024

Dear Dr. DuBois,

We are pleased to inform you that your manuscript 'Structure and antigenicity of the divergent human astrovirus VA1 capsid spike' has been provisionally accepted for publication in PLOS Pathogens.

Best regards,

Félix A. Rey

Academic Editor

PLOS Pathogens

Guangxiang Luo

Section Editor

PLOS Pathogens

Michael Malim

Editor-in-Chief

PLOS Pathogens

orcid.org/0000-0002-7699-2064
---

## [Editor Report · Acceptance letter]

22 Feb 2024

Dear Dr. DuBois,

We are delighted to inform you that your manuscript, "Structure and antigenicity of the divergent human astrovirus VA1 capsid spike," has been formally accepted for publication in PLOS Pathogens.

Best regards,

Michael Malim

Editor-in-Chief

PLOS Pathogens

orcid.org/0000-0002-7699-2064